# Ethnic disparities in initiation and intensification of diabetes treatment in adults with type 2 diabetes in the UK, 1990–2017: A cohort study

Rohini Mathur[1]*, Ruth E. Farmer[1], Sophie V. Eastwood[2], Nish Chaturvedi[2], Ian Douglas[1], Liam Smeeth[1]

1 Department of Non-communicable Disease Epidemiology, London School of Hygiene & Tropical Medicine, London, United Kingdom, 2 Institute of Cardiovascular Sciences, University College London, London, United Kingdom

* rohini.mathur@lshtm.ac.uk

**Data Availability Statement:** Data were obtained from the CPRD (www.cprd.com). CPRD is a research service that provides primary care and

## Abstract

### Background

Type 2 diabetes mellitus (T2DM) disproportionately affects individuals of nonwhite ethnic origin. Timely and appropriate initiation and intensification of glucose-lowering therapy is key to reducing the risk of major vascular outcomes. Given that ethnic inequalities in outcomes may stem from differences in therapeutic management, the aim of this study was to identify ethnic differences in the timeliness of initiation and intensification of glucose-lowering therapy in individuals newly diagnosed with T2DM in the United Kingdom.

### Methods and findings

An observational cohort study using the Clinical Practice Research Datalink was conducted using 162,238 adults aged 18 and over diagnosed with T2DM between 1990 and 2017 (mean age 62.7 years, 55.2% male); 93% were of white ethnicity ($n = 150,754$), 5% were South Asian ($n = 8,139$), and 2.1% were black ($n = 3,345$). Ethnic differences in time to initiation and intensification of diabetes treatment were estimated at three time points (initiation of noninsulin monotherapy, intensification to noninsulin combination therapy, and intensification to insulin therapy) using multivariable Cox proportional hazards regression adjusted for factors a priori hypothesised to be associated with initiation and intensification: age, sex, deprivation, glycated haemoglobin (HbA1c), body mass index (BMI), smoking status, comorbidities, consultations, medications, calendar year, and clustering by practice. Odds of experiencing therapeutic inertia (failure to intensify treatment within 12 months of HbA1c >7.5% [58 mmol/mol]), were estimated using multivariable logistic regression adjusted for the same hypothesised confounders. Noninsulin monotherapy was initiated earlier in South Asian and black groups (South Asian HR 1.21, 95% CI 1.08–1.36, $p < 0.001$; black HR 1.29, 95% CI 1.05–1.59, $p = 0.017$). Correspondingly, no ethnic differences in therapeutic inertia were evident at initiation. Intensification with noninsulin combination therapy was slower in both nonwhite ethnic groups relative to white (South Asian HR 0.80, 95% CI 0.74–0.87, $p <$

linked data for public health research. CPRD data governance and our own licence to use CPRD data do not allow us to distribute or make available patient data directly to other parties. Researchers can apply for data access with CPRD and must have their study protocol approved by the Independent Scientific Advisory Committee for Medicines and Healthcare products Regulatory Agency database research.

**Funding:** RM is funded by a Sir Henry Wellcome Postdoctoral Fellowship (201375/Z/16/Z, www. wellcome.ac.uk). The funders had no role in study design, data collection and analysis, decision to publish, or preparation of the manuscript.

**Competing interests:** I have read the journal's policy and the authors of this manuscript have the following competing interests: ID holds grants from GSK, ABPI, NIHR, and MRC outside the submitted work and is a GSK shareholder. REF is now an employee of Boehringer Ingelheim; her involvement in the submitted work is independent of the role.

**Abbreviations:** BMI, body mass index; CPRD, Clinical Practice Research Datalink; DBP, diastolic blood pressure; DPP4i, dipeptidyl peptidase-4 inhibitor; EHR, electronic health record; GLP-1, glucagon-like peptide-1; HbA1c, glycated haemoglobin; IFCC, International Federation of Clinical Chemistry; IMD, Index of Multiple Deprivation; ISAC, Independent Scientific Advisory Committee; NIAD, noninsulin antidiabetic drug; OR, odds ratio; SBP, systolic blood pressure; SGLT2i, sodium-glucose cotransporter-2 inhibitor; T2DM, type 2 diabetes mellitus.

0.001; black HR 0.79, 95% CI 0.70–0.90, $p < 0.001$); treatment inertia at this stage was greater in nonwhite groups relative to white (South Asian odds ratio [OR] 1.45, 95% CI 1.23–1.70, $p < 0.001$; black OR 1.43, 95% CI 1.09–1.87, $p = 0.010$). Intensification to insulin therapy was slower again for black groups relative to white groups (South Asian HR 0.49, 95% CI 0.41–0.58, $p < 0.001$; black HR 0.69, 95% CI 0.53–0.89, $p = 0.012$); correspondingly, treatment inertia was significantly higher in nonwhite groups at this stage relative to white groups (South Asian OR 2.68, 95% CI 1.89–3.80 $p < 0.001$; black OR 1.82, 95% CI 1.13–2.79, $p = 0.013$). At both stages of treatment intensification, nonwhite groups had fewer HbA1c measurements than white groups. Limitations included variable quality and completeness of routinely recorded data and a lack of information on medication adherence.

## Conclusions

In this large UK cohort, we found persuasive evidence that South Asian and black groups intensified to noninsulin combination therapy and insulin therapy more slowly than white groups and experienced greater therapeutic inertia following identification of uncontrolled HbA1c. Reasons for delays are multifactorial and may, in part, be related to poorer long-term monitoring of risk factors in nonwhite groups. Initiatives to improve timely and appropriate intensification of diabetes treatment are key to reducing disparities in downstream vascular outcomes in these populations.

## Author summary

### Why was this study done?

- In the UK, ethnic minority populations, particularly of South Asian and black African/Caribbean descent, have a higher risk of type 2 diabetes mellitus (T2DM) and related adverse outcomes, such as cardiovascular disease, than the white population.
- Timely and appropriate diabetes treatment can substantially reduce risk of adverse outcomes associated with T2DM.
- We sought to quantify ethnic differences in time to initiation and intensification of diabetes treatment among individuals newly diagnosed with T2DM to assess whether these clinically modifiable factors may contribute to ethnic differences in outcomes.

### What did the researchers do and find?

- We used routinely recorded data from general practices across the UK to identify people newly diagnosed with T2DM and compared how long it took to initiate and intensify diabetes treatment, comparing people of white, South Asian, and black ethnicity.
- We found that South Asian and black groups initiated diabetes treatment more quickly than white groups but were slower to intensify to second- and third-line treatment regimes.

**What do these findings mean?**

- Although time to initial treatment of type 2 diabetes was appropriate, ethnic disparities in subsequent longer-term treatment may contribute to the worse outcomes seen in ethnic minority populations in the UK.

- Interventions to improve timely and appropriate intensification of diabetes treatment are key to reducing disparities in the downstream adverse outcomes of T2DM.

## Introduction

The burden of type 2 diabetes mellitus (T2DM) is growing worldwide, disproportionately affecting individuals of nonwhite ethnic origin [1–3]. Around 6% of the UK population have T2DM, with the risk of developing T2DM approximately 2- to 4-fold greater in migrant populations of South Asian or black African or Caribbean descent. Additionally, once diagnosed with T2DM, the lifetime risk of developing cardiovascular conditions is higher again in minority ethnic groups than those of white European origin [4–11].

Timely and appropriate initiation and intensification of glucose-lowering therapy have been shown to substantively reduce adverse diabetes outcomes [12,13]. In the UK, clinical guidelines for the therapeutic management of blood glucose recommend that newly diagnosed individuals initiate with noninsulin monotherapy, usually metformin, if their glycated haemoglobin (HbA1c) level is above 6.5% (48 mmol/mol). Intensification with additional noninsulin therapies, and ultimately insulin, is recommended if HbA1c remains above 7.5% (58 mmol/mol) [14].

Poor glycaemic control is associated with an increased risk of macrovascular complications (such as coronary disease and stroke) and microvascular complications (such as diabetic retinopathy, neuropathy, and kidney disease). One key driver of suboptimal glucose management is therapeutic inertia, defined as failure to appropriately initiate or intensify treatment in a timely manner following identification of uncontrolled risk factor levels. In the UK, it has been shown that one-third of individuals do not achieve target HbA1c levels of 7.5% (58 mmol/mol), despite clinical guidelines recommending that therapy be intensified within 3–6 months of identifying a raised HbA1c [14,15]. Furthermore, a 2019 UK study has highlighted the high economic burden associated with therapeutic inertia, with costs related to increases in diabetes-related complications and lost workplace productivity equalling £2.6 million over the next 10 years [16].

Given that ethnic inequalities in diabetes care and outcomes can accumulate from even before diagnosis, it is essential to identify where along the care pathway disparities arise. In particular, it is essential to identify easily modifiable aspects of the pathway that may contribute to inequalities. As such, the aim of this study is to identify ethnic differences in the timeliness of initiation and intensification of glucose-lowering treatment in individuals with newly diagnosed T2DM in the UK—specifically, (1) ethnic differences in the ordering of glucose-lowering treatments, (2) ethnic differences in time to initiation and intensification of treatment, and (3) ethnic differences in treatment delays (therapeutic inertia) following identification of uncontrolled HbA1c.

## Methods

### Study design and population

An observational cohort study utilising the Clinical Practice Research Datalink (CPRD) was undertaken. The CPRD is a clinical research database containing anonymised longitudinal

primary care records for approximately 15 million people from 714 general practices and has been shown to be representative of the UK population with respect to age, sex, and ethnicity [17]. Adults aged 18 and over who were registered between January 1990 and December 2017, with at least 12 months of continuous registration prior to first recorded diagnosis of T2DM, were included in the study. T2DM was identified using an adjudication algorithm developed to minimise misclassification of diabetes status and type in electronic health records (EHRs) [18].

Transitions between treatment stages were analysed in three time periods: (1) time from diagnosis to initiation of noninsulin monotherapy, (2) time from noninsulin monotherapy to intensification with noninsulin combination therapy, and (3) time from noninsulin combination therapy to intensification with insulin therapy.

### Covariates

Self-reported ethnicity was identified using Read codes and collapsed into the five main categories of the 2001 UK census (white, South Asian, black African/Caribbean, mixed, and other). For individuals with more than one ethnicity code on their primary care record, an algorithm was used to assign a best 'single' ethnicity—based on the most commonly and most recently recorded codes (S1 Fig) [19]. Age at diagnosis was calculated by subtracting the year of birth from year of diagnosis and was divided into 10-year age bands. Deprivation was measured using quintiles of the Index of Multiple Deprivation (IMD) [20].

For the time from diagnosis to initiation with noninsulin monotherapy, baseline was defined as the date of T2DM diagnosis. Baseline covariates were identified from the value closest to the date of T2DM diagnosis from the 12 months preceding or the 3 months following diagnosis. These included HbA1c, body mass index (BMI), systolic blood pressure (SBP) and diastolic blood pressure (DBP), and smoking status (categorised as 'never smoker', 'current smoker', and 'ex-smoker'). Although BMI was considered as a continuous covariate in the analytical models, descriptive statistics categorised BMI according to the standard categories of underweight, normal weight, overweight, and obese, with adjustments for South Asian ethnicity as recommended by WHO, which classify normal weight as between 18.5 and 22.9 kg/m$^2$, overweight as between 23 and 27.4 kg/m$^2$, and obesity as 27.5 kg/m$^2$ or over for this ethnic group [21].

Comorbidities were considered present at baseline if recorded at any time prior to diagnosis. These included depression, macrovascular disease (hypertension, myocardial infarction, angina, stroke, and heart failure), and microvascular disease (chronic kidney disease, retinopathy, and neuropathy) (see S1 Table for all code lists). The number of consultations and oral medications in the 6 months prior to diagnosis was also included as baseline covariates.

For the time from noninsulin monotherapy to noninsulin combination therapy, baseline was defined as the date of initiation of noninsulin therapy. Baseline HbA1c, BMI, and smoking status were derived from the date closest to the date of monotherapy initiation in the 6 months preceding. Counts of consultations and oral medications were calculated from the 6 months prior to monotherapy initiation. Comorbidities were considered present if recorded at any time prior to monotherapy initiation. For the time from noninsulin combination therapy to insulin therapy, baseline was defined as the date of initiation of noninsulin combination therapy, with baseline HbA1c, BMI, smoking status, counts of consultations and oral medications, and comorbidities defined as above.

For each of the three time periods of interest, additional between-treatment variables were constructed. These included the number of HbA1c measurements and the number of consultations between treatment stages.

## Glucose-lowering treatment

Glucose-lowering treatment was identified from GP prescribing data and categorised into six classes: metformin, sulfonylurea, newer agents (dipeptidyl peptidase-4 inhibitor [DPP4i], sodium-glucose cotransporter-2 inhibitor [SGLT2i], and glucagon-like peptide-1 [GLP-1] receptor agonists), thiazolidinediones, insulin, and other drugs. Newer agents were grouped together because they represent treatment strategies used for similar stages of diabetes progression.

## Statistical analysis

**Patterns of glucose-lowering treatment.**   Firstly, the proportion of individuals prescribed each class of glucose-lowering drug was calculated from the initiation of a single drug through to the addition of up to four more drugs and compared between ethnic groups. Secondly, patterns of treatment up-titration were compared between ethnic groups using sequence analysis. Sequences included (1) staying on a single drug regime for the entire follow-up period, (2) moving between two drug regimes, and (3) moving between three drug regimes.

## Time to treatment initiation and intensification

Multilevel multivariable proportional hazard models assuming an exponential baseline hazard function were employed to identify ethnic differences in time to initiation and intensification of glucose-lowering treatment while accounting for the clustering of individuals within general practices. Individuals free from glucose-lowering treatment at the date of T2DM diagnosis were eligible for inclusion. Individuals who initiated diabetes treatment in the 90 days prior to diagnosis were considered to be 'baseline initiators' and, for analytical purposes, had their initiation date moved to 1 day after diagnosis to allow entry into the cohort. For initiation of non-insulin monotherapy, follow-up time began at the date of T2DM diagnosis and ended at the date of initiation. For intensification to noninsulin combination therapy, follow-up time began at commencement of monotherapy and ended at the date of intensification with combination therapy. For intensification to insulin therapy, follow-up time began at the commencement of noninsulin combination therapy and ended at the date of intensification with insulin.

For individuals who did not initiate or intensify with the drug of interest during the follow-up period, follow-up time was censored at the earliest of the following: date of intensifying to a different drug, death, transferring out of the practice, or last data collection. For example, individuals who initiated with a treatment other than noninsulin monotherapy were censored at the date the alternative treatment was commenced, and individuals who intensified directly from noninsulin monotherapy to insulin were censored at the date of insulin initiation.

All models adjusted for hypothesised covariates as identified in the directed acyclic diagram (S2 Fig); namely, age at diagnosis, sex, deprivation, year of diabetes diagnosis (to account for secular trends in prescribing guidelines and treatment availability), HbA1c, BMI, and smoking status at the start of each follow-up period; presence of depression, micro- and macrovascular comorbidities at the start of each follow-up period; and count of consultations and oral medications in the 6 months prior to the start of each follow-up period. Models for intensification to combination therapy and insulin therapy additionally adjusted for time since diagnosis. As individuals attending the same general practice may have similar levels of care provision and clinical coding, multilevel modelling was used to account for the clustering of people within practices.

Because of overdispersion in the Poisson model, multilevel multivariable negative binomial regression was used to estimate ethnic differences in the count of HbA1c measurements, and

consultations between treatment stages adjusted for all hypothesised confounders and clustering by practice.

## Ethnic differences in therapeutic inertia

As there is no single accepted definition of therapeutic inertia, we adopted a definition used in previous UK database studies [22–24]. For the purposes of our study, therapeutic inertia was defined as the failure to intensify treatment within 12 months of having an HbA1c of >7.5% (58 mmol/mol) recorded by the general practitioner. Individuals with any raised HbA1c following (1) diagnosis, (2) initiation of noninsulin monotherapy, or (3) initiation of noninsulin combination therapy with at least 12 months of follow-up were included in the analysis. Models to estimate ethnic differences in the odds of experiencing therapeutic inertia were constructed adjusting for all hypothesised confounders. Models for treatment inertia around time of intensification to combination therapy and insulin therapy additionally adjusted for time between first raised HbA1c and T2DM diagnosis in each of the three time periods. Descriptive statistics for factors associated with therapeutic inertia including the number of HbA1c's above 7.5% (58 mmol/mol) measured between first raised HbA1c and treatment intensification, the number of consultations between first raised HbA1c and treatment intensification, and the proportion of individuals with a consultation within 3 months of the first recorded raised HbA1c were calculated and presented by ethnic group.

For all analyses, individuals of white ethnicity were considered as the reference population. Comparisons between the white, South Asian, and black African/Caribbean populations are reported in the main results. Analyses were conducted as specified in the scientific protocol (see S1 Text). Due to the fact that routinely recorded data in primary care EHRs are likely to be missing not at random, multiple imputation of missing data was not considered appropriate because the assumption of missing at random was unlikely to be met. As such, a complete case analysis approach was employed [25]. All analyses were completed using Stata version 15 and reported according to the RECORD guidelines (see S1 RECORD Checklist).

## Secondary analysis

Firstly, as current guidelines for diabetes management in the UK suggest a threshold of 6.5% (48 mmol/mol) for initiation of diabetes treatment, a secondary analysis of ethnic differences in treatment inertia at initiation of noninsulin monotherapy was conducted using >6.5% at the definition of raised HbA1c instead of >7.5% (58 mmol/mol). Secondly, as 12 months' minimum follow-up was required for inclusion into the analysis of treatment inertia, the ethnic breakdown of those included in the analysis was compared to that of those excluded for lack of 12 months' follow-up at each stage.

## Ethical approval

Ethical and scientific approval for this study were granted by the Independent Scientific Advisory Committee (ISAC) (protocol 17_087R) and the London School of Hygiene & Tropical Medicine (project ID 13409).

## Results

From a total of 425,811 adults aged 18 and over who were diagnosed with T2DM between 1990 and 2017, 162,238 individuals of white, South Asian, or black ethnicity with at least 1 year of continuous registration prior to diagnosis were included in the study (Fig 1). Individuals of mixed/other ethnicity (*n* = 2,794) and unknown ethnicity (*n* = 75,258) were excluded from the

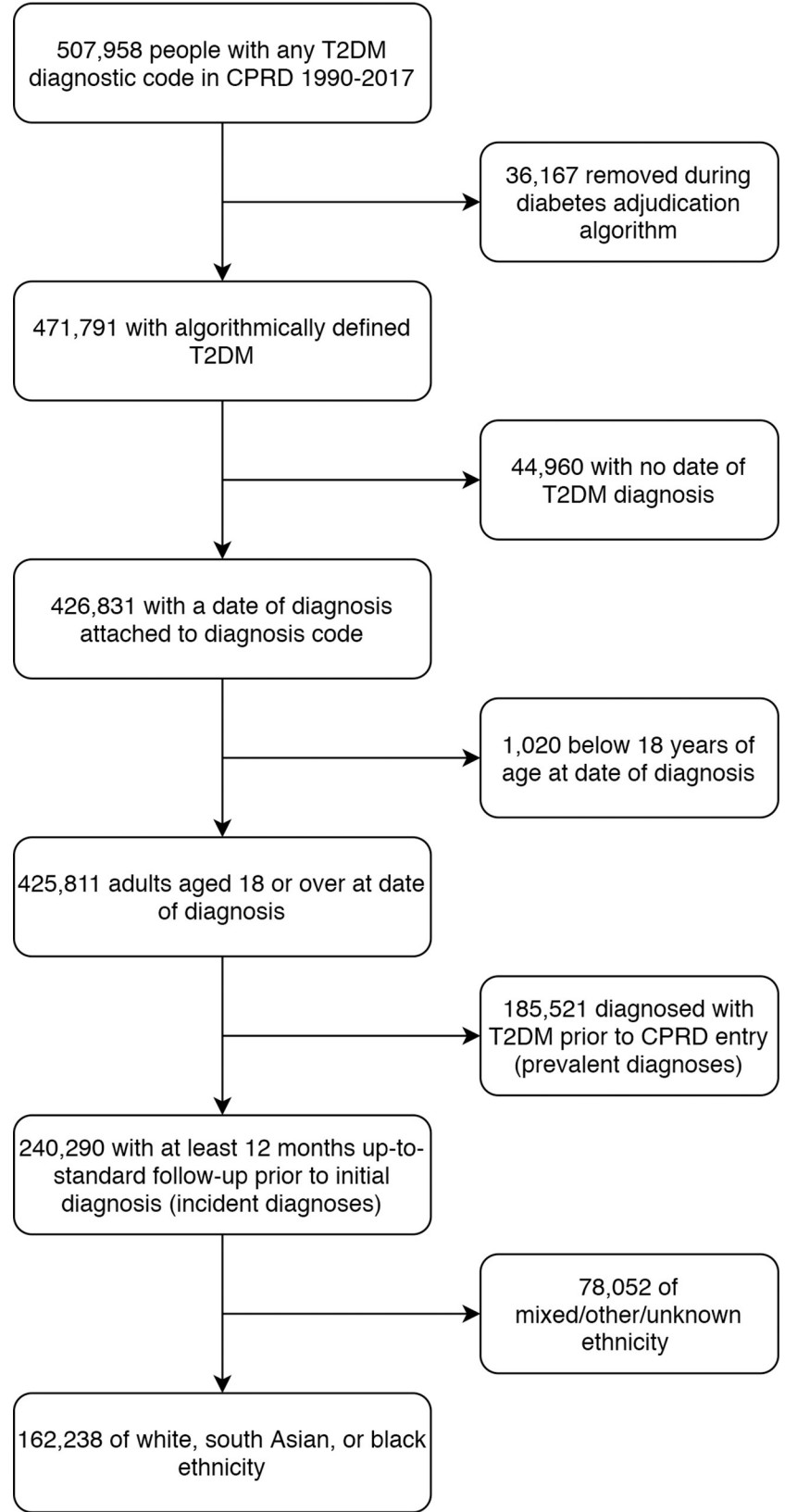

**Fig 1. Population inclusion flowchart.** CPRD, Clinical Practice Research Datalink; T2DM, type 2 diabetes mellitus.

study population (comparisons between the white, other, and unknown ethnic groups are reported in S2–S4 Tables). Individuals in the study cohort contributed a mean of 6.2 years of follow-up. The ethnic breakdown of the study cohort was 92.9% white (*n* = 150,754), 5.0% South Asian (*n* = 8,139), and 2.1% black (*n* = 3,345). Compared with the white group, age at diagnosis was markedly younger in ethnic minority groups (South Asian, 55 years; black, 56 years; white, 63 years), and baseline HbA1c was higher (South Asian, 65.8 mmol/mol; black, 69.3 mmol/mol; white, 64 mmol/mol) (Table 1).

## Patterns of glucose-lowering treatment

Overall, 36.2% of individuals remained on noninsulin monotherapy for the entire study period, 26.0% intensified from noninsulin monotherapy to noninsulin combination therapy, and 6.8% intensified from monotherapy to combination therapy to insulin therapy. Individuals with treatment patterns differing from guidelines included those who initiated with noninsulin combination therapy (3.5%) or insulin therapy (1.2%), those who intensified from noninsulin monotherapy directly to insulin (1.8%), and those who initiated with combination therapy and intensified to insulin (1.0%); 23.8% of the study population did not initiate any glucose-lowering treatment during the study period (Fig 2).

The ordering of glucose-lowering drug classes was comparable between ethnic groups at initiation but diverged in later stages. In all ethnic groups, the most popular first-line treatment was metformin (82%), the most popular second-line treatment was sulfonylureas (50%), and the most popular third-line treatments were newer-generation agents (34%). The choice for the fourth and fifth additional agent differed substantially by ethnic group; whereas South Asian and black groups were most likely to receive an additional newer-generation agent in both stages, white groups were more likely to receive insulin (Fig 3).

## Time to treatment initiation and intensification

Median time to initiation of noninsulin monotherapy was 3.2 months. Median time to intensification to noninsulin combination therapy was 28.5 months (2.4 years), and median time to intensification to insulin therapy was 45 months (3.8 years).

After adjusting for age, sex, deprivation, comorbidities, baseline HbA1c, BMI and smoking status, count of consultations and medications, calendar year, and clustering by practice, time to initiation was 21% faster in South Asian groups (95% CI 8%–36%, *p* < 0.001) and 29% faster in black groups relative to white (95% CI 5%–59%, *p* = 0.017). In contrast, time to intensification with noninsulin combination therapy was significantly slower for both nonwhite ethnic groups relative to white (South Asian HR 0.80, 95% CI 0.74–0.87, *p* < 0.001; black HR 0.79, 95% CI 0.70–0.90, *p* < 0.001). Ethnic differences widened further for intensification to insulin therapy, with South Asian groups taking twice as long to intensify as white groups (HR 0.49, 95% CI 0.41–0.58, *p* < 0.001) and black groups taking 31% longer to intensify (HR 0.69, 95% CI 0.53–0.89, *p* < 0.001) (Table 2, S3 Fig).

In all treatment stages, nonwhite ethnic groups had fewer HbA1c measurements than white groups after adjusting for all confounders (Table 2).

## Ethnic differences in therapeutic inertia

From the total study population, 79,720 individuals with at least 12 months of follow-up following their first raised HbA1c value of >7.5% (58 mmol/mol) were included in the analysis of treatment inertia (S4 Fig).

Of all individuals who were treatment naive at diagnosis, 18% experienced therapeutic inertia when initiating treatment with noninsulin monotherapy. After accounting for all

**Table 1. Baseline characteristics stratified by ethnic group.**

| Baseline characteristics | White | | SA | | Black | |
|---|---|---|---|---|---|---|
| N | 150,754 | | 8,139 | | 3,345 | |
| Years of follow-up (mean, SD) | 6.3 | (4.6) | 5.9 | (4.7) | 5.1 | (4.4) |
| Age at diagnosis (mean, SD) | 63.4 | (13.2) | 53 | (12.9) | 55.8 | (13) |
| Gender, male, n (%) | 82,619 | (54.8) | 4,411 | (54.2) | 1,679 | (50.2) |
| Deprivation quintile | | | | | | |
| 1 (least deprived), n (%) | 27,606 | (18.3) | 1,030 | (12.7) | 191 | (5.7) |
| 2, n (%) | 30,092 | (20) | 1,321 | (16.2) | 306 | (9.1) |
| 3, n (%) | 33,318 | (22.1) | 1,721 | (21.1) | 717 | (21.4) |
| 4, n (%) | 28,989 | (19.2) | 1,803 | (22.2) | 929 | (27.8) |
| 5 (most deprived), n (%) | 30,749 | (20.4) | 2,264 | (27.8) | 1,202 | (35.9) |
| Smoking status | | | | | | |
| Never smoker, n (%) | 51,565 | (34.2) | 4,619 | (56.8) | 1,721 | (51.4) |
| Current smoker, n (%) | 23,503 | (15.6) | 843 | (10.4) | 342 | (10.2) |
| Ex-smoker, n (%) | 47,629 | (31.6) | 828 | (10.2) | 488 | (14.6) |
| Missing, n (%) | 28,057 | (18.6) | 1,849 | (22.7) | 794 | (23.7) |
| BMI | | | | | | |
| BMI at diagnosis, kg/m$^2$ (mean, SD) | 31.7 | (6.1) | 29.2 | (5.2) | 31.3 | (5.9) |
| Underweight (<20, <18.5 for SA) | 1,447 | (1) | 23 | (.3) | 24 | (.7) |
| Normal weight (20–25, 18.4–23 for SA) | 13,560 | (9) | 532 | (6.6) | 309 | (9.2) |
| Overweight (25–30, 23.5–27.5 for SA) | 39,826 | (26.4) | 2,251 | (27.8) | 917 | (27.4) |
| Obese (>30, >27.5 for SA) | 71,664 | (47.5) | 4,013 | (49.5) | 1,521 | (45.5) |
| Missing | 24,257 | (16.1) | 1,287 | (15.9) | 574 | (17.2) |
| HbA1c | | | | | | |
| HbA1c at diagnosis % (mean, SD) | 8 | (2.1) | 8.2 | (2.1) | 8.5 | (2.4) |
| HbA1c at diagnosis, IFCC (mean, SD) | 63.6 | (23.2) | 65.8 | (22.7) | 69.3 | (26.7) |
| ≤7.5%, n (%) | 64,642 | (42.9) | 3,285 | (40.4) | 1,284 | (38.4) |
| 7.5%–7.9%, n (%) | 11,191 | (7.4) | 767 | (9.4) | 305 | (9.1) |
| 8.0%–8.9%, n (%) | 13,291 | (8.8) | 847 | (10.4) | 327 | (9.8) |
| ≥9.0%, n (%) | 29,594 | (19.6) | 1,692 | (20.8) | 849 | (25.4) |
| Missing, n (%) | 32,036 | (21.3) | 1,548 | (19) | 580 | (17.3) |
| Blood pressure | | | | | | |
| SBP at diagnosis (mean, SD) | 140.3 | (18.8) | 133.2 | (17.6) | 137.5 | (18.5) |
| DBP at diagnosis (mean, SD) | 80.9 | (10.9) | 80.8 | (10.5) | 82.5 | (10.8) |
| <140/90, n (%) | 27,185 | (19) | 1,131 | (14.8) | 671 | (21.1) |
| <150/90, n (%) | 20,424 | (14.3) | 737 | (9.6) | 471 | (14.8) |
| <130/80, n (%) | 76,154 | (53.2) | 3,384 | (44.3) | 1,683 | (52.9) |
| Missing, n (%) | 7,564 | (5) | 500 | (6.1) | 166 | (5) |
| Comorbidities and medications | | | | | | |
| Any macrovascular, n (%) | 21,669 | (14.4) | 685 | (8.4) | 198 | (5.9) |
| Any microvascular, n (%) | 4,685 | (3.1) | 202 | (2.5) | 96 | (2.9) |
| Depression, n (%) | 34,246 | (22.7) | 1973 | (24.2) | 721 | (21.6) |
| On antihypertensive at diagnosis, n (%) | 43,427 | (28.8) | 1,428 | (17.5) | 555 | (16.6) |
| On statin at diagnosis, n (%) | 77,072 | (51.1) | 3,641 | (44.7) | 1,295 | (38.7) |
| DM treatment initiation characteristics | | | | | | |
| Initiate <1 year before diagnosis, n (%) | 10,654 | (7.1) | 691 | (8.5) | 282 | (8.4) |
| Initiate in 12 months prior to diagnosis, n (%) | 33,034 | (21.9) | 2,347 | (28.8) | 1,086 | (32.5) |
| Initiate within 90 days of diagnosis, n (%) | 27,804 | (18.4) | 1,609 | (19.8) | 704 | (21) |

*(Continued)*

**Table 1.** (Continued)

| Baseline characteristics | White | | SA | | Black | |
|---|---|---|---|---|---|---|
| Initiate >90 days after diagnosis, n (%) | 45,187 | (30) | 2,293 | (28.2) | 666 | (19.9) |
| Noninitiators of DM treatment, n (%) | 34,075 | (22.6) | 1,199 | (14.7) | 607 | (18.1) |

Baseline measures of HbA1c, BP, and CVD risk defined as value closest to diagnosis date in the 12 months prior or 3 months after.

Abbreviations: BMI, body mass index; BP, blood pressure; CVD, cardiovascular disease; DBP, diastolic blood pressure; DM, diabetes mellitus; HbA1c, glycated haemoglobin; IFCC, International Federation of Clinical Chemistry; SA, South Asian; SBP, systolic blood pressure

hypothesised confounders, no ethnic differences in therapeutic inertia were evident (South Asian odds ratio [OR] 0.97, 95% CI 0.76–1.24, $p$ = 0.796; black OR 0.94, 95% CI 0.69–1.28, $p$ = 0.683).

Of all individuals on noninsulin monotherapy, 68% experienced therapeutic inertia when intensifying to noninsulin combination therapy. Both South Asian and black groups experienced greater therapeutic inertia than white groups at this stage (South Asian OR 1.45, 95% CI 1.23–1.70, $p$ < 0.001; black OR 1.43, 95% CI 1.09–1.87, $p$ = 0.010).

Over 93% of all individuals on noninsulin combination therapy experienced therapeutic inertia when intensifying to insulin therapy. At this stage, ethnic minority groups were substantially more likely to experience therapeutic inertia than white groups (South Asian OR 2.68, 95% CI 1.89–3.80, $p$ < 0.001; black OR 1.82, 95% CI 1.13–2.92, $p$ = 0.013) (Table 3).

## Secondary analyses

As in the primary analysis, no ethnic differences in the odds of experiencing therapeutic inertia were evident for initiation of noninsulin monotherapy when using 6.5% (48 mmol/mol) as the cutoff for raised HbA1c instead of >7.5% (58 mmol/mol) (South Asian OR 0.86, 95% CI 0.67–1.12, $p$ = 0.274; black OR 0.86, 95% CI 0.60–1.23, $p$ = 405) (S5 Table). Furthermore, the ethnic

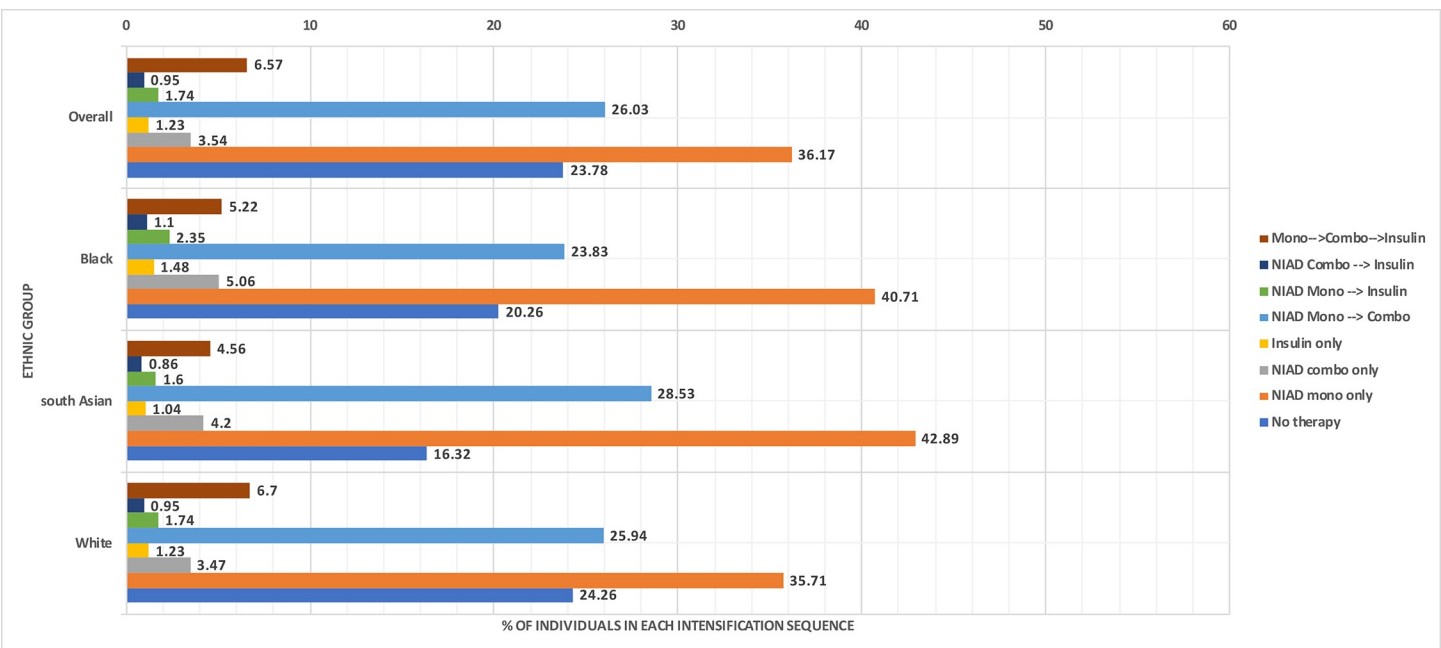

**Fig 2. Diabetes therapy intensification sequence from diagnosis to end of follow-up by ethnic group.** NIAD, noninsulin antidiabetic drug.

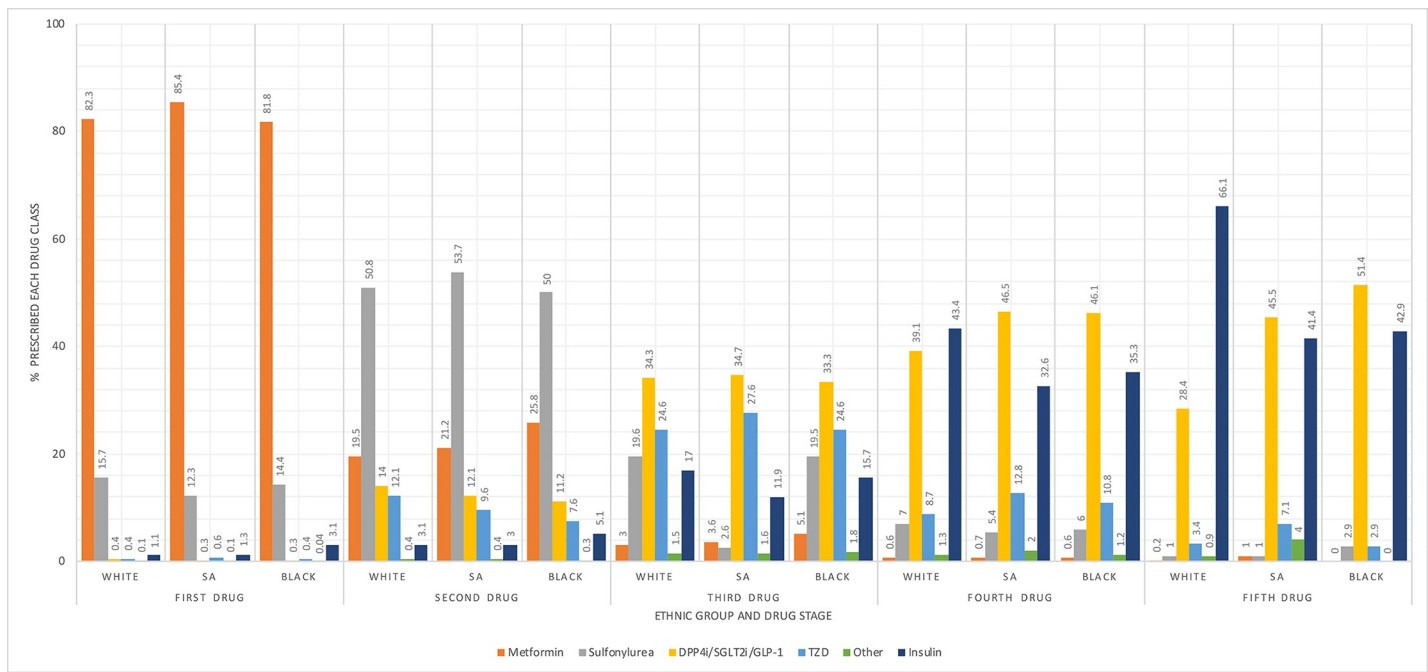

**Fig 3. Drug classes by intensification stage and ethnic group.** DPP4i, dipeptidyl peptidase-4 inhibitor; GLP-1, glucagon-like peptide-1; SA, South Asian; TZD, thiazolidinedione.

breakdown of individuals eligible for inclusion in the analysis of therapeutic inertia was comparable to the ethnic breakdown of individuals excluded due to insufficient follow-up after their initial raised HbA1c measure (S6 Table, full models for all analyses are in S7 Table).

## Discussion

To our knowledge, this is the first UK-based study to examine the extent to which the appropriate and timely prescribing of diabetes treatment among individuals with type 2 diabetes differs by ethnic group. We report two main findings: Firstly, despite initiating noninsulin monotherapy more quickly after diagnosis than white groups, South Asian and black groups were more likely to experience delays when intensifying to noninsulin combination therapy and insulin therapy. Secondly, upon having uncontrolled HbA1c identified by the healthcare provider, South Asian and black groups were more likely to experience therapeutic inertia when intensifying to combination and insulin therapy.

Although several previous studies have quantified timeliness of initiation and intensification of antidiabetic medication in UK and international populations, ours is the first to identify differences by ethnic group. Similarly, although the literature describing ethnic differences in diabetic outcomes is extensive, we propose a mechanism via which these inequalities might occur—namely, via delayed intensification of treatment and therapeutic inertia following the identification of uncontrolled risk. Recognising this mechanism enables us to identify several priority target areas for reducing ethnic inequalities in the long-term management of diabetes in primary care settings.

Our data suggest that excessive delays in treatment intensification in ethnic minority populations may result from poorer monitoring of risk factors in these populations; in our study, both South Asian and black groups received fewer HbA1c measurements than white groups prior to intensification with both noninsulin combination therapy and insulin.

**Table 2. Time to antidiabetic treatment initiation and intensification.**

| Initiation and intensification characteristics | Initiation of noninsulin monotherapy | | | Intensification to noninsulin combination therapy | | | Intensification to insulin therapy | | |
|---|---|---|---|---|---|---|---|---|---|
| *N* eligible to initiate/intensify | | 146,693 | | | 113,518 | | | 60,108 | |
| | **White** | **South Asian** | **Black** | **White** | **South Asian** | **Black** | **White** | **South Asian** | **Black** |
| *N* eligible to initiate/intensify | 136,540 | 7,247 | 2,906 | 105,025 | 6,224 | 2,269 | 55,872 | 3,097 | 1,139 |
| Percent who initiate/intensify at any time | 98.9% | 99.4% | 97.8% | 46.2% | 42.8% | 39.0% | 20.9% | 14.1% | 16.2% |
| Duration of diabetes at start of follow-up period (years, mean, SD) | 0 | 0 | 0 | 1.00 (1.99) | 0.72 (1.63) | 0.54 (1.47) | 2.97 (3.05) | 2.90 (3.05) | 2.30 (2.92) |
| **Time to treatment initiation/ intensification** | | | | | | | | | |
| Months to initiation/intensification (mean, SD) | 6.2 (33.5) | 2.3 (17.4) | 1.2 (15.4) | 29.6 (45.1) | 29.3 (43.6) | 26.7 (42.6) | 45.5 (59.8) | 47.5 (64.5) | 42.7 (60.6) |
| Relative risk versus white (HR, 95% CI, *p*-value) | 1 | 1.21 (1.08–1.36) <0.001 | 1.29 (1.05–1.59) 0.017 | 1 | 0.80 (0.74–0.87) <0.001 | 0.79 (0.70–0.90) <0.001 | 1 | 0.49 (0.41–0.58) <0.001 | 0.69 (0.53–0.89) 0.004 |
| **Between-treatment characteristics** | | | | | | | | | |
| Mean HbA1c at diagnosis/start of follow-up period (%) | 8 (2.1) | 8.1 (2) | 8.4 (2.4) | 8.6 (1.9) | 8.5 (1.9) | 8.9 (2.3) | 8.9 (1.8) | 9 (1.9) | 9.6 (2.3) |
| Mean HbA1c closest to date of initiation/intensification (%) | 8.6 (2) | 8.6 (1.9) | 8.9 (2.3) | 8.8 (1.7) | 8.9 (1.8) | 9.3 (2.1) | 10 (1.9) | 10.1 (1.9) | 11 (2.5) |
| Number of HbA1c measurements between treatment stages (mean, SD) | 1.5 (2.9) | 1.1 (2.3) | 0.8 (2.1) | 1.7 (3.1) | 1.2 (2.3) | .9 (2.1) | 1.4 (2.5) | 1 (1.9) | 0.8 (2.1) |
| HbA1c count (RR, 95% CI, *p*-value) | 1 | 0.94 (0.90–0.98) 0.002 | 0.90 (0.83–0.98) 0.017 | 1 | 0.78 (0.70–0.85) <0.001 | 0.72 (0.60–0.87) <0.001 | 1 | 0.74 (0.63–0.87) <0.001 | 0.64 (0.50–0.82) <0.001 |
| Number of consultations between treatment stages (mean, SD) | 12 (27.6) | 9 (20.7) | 6.9 (19) | 14.8 (29.9) | 10 (21.4) | 8 (20.6) | 11.3 (23.5) | 7.9 (17.8) | 6.4 (16.4) |
| Consultation (RR, 95% CI, *p*-value) | 1 | 0.98 (0.94–1.02) 0.329 | 0.98 (0.93–1.04) 0.541 | 1 | 0.89 (0.78–1.01) 0.071 | 0.82 (0.66–1.01) 0.062 | 1 | 0.63 (0.52–0.76) <0.001 | 0.77 (0.59–1.01) 0.061 |

Population eligible for initiation excludes those on any diabetes drug in 90 days prior to diagnosis. All models adjusted for age, sex, deprivation, year of diagnosis, HbA1c, BMI, micro- and macro vascular comorbidities, depression, consultation count, smoking status and medication count at start of follow-up period, and clustering by practice. Models for intensification to combination therapy and insulin additionally account for time since diagnosis. Mean HbA1c and BMI taken as the latest in the 6 months prior to diagnosis (for model 1), initiation (for model 2), and intensification 1 (for model 3).

Abbreviations: BMI, body mass index; HbA1c, glycated haemoglobin; RR, rate ratio

A recent systematic review of 53 studies worldwide has highlighted the widespread problem of therapeutic inertia in diabetes—with delays common across all stages of treatment initiation and intensification, though most pronounced around the time of intensification to insulin [22,26]. Correspondingly, we found that 67% of those intensifying to noninsulin combination therapy and 93% of those intensifying to insulin therapy experienced treatment inertia.

## Strengths and weaknesses of this study

The strengths and limitations of routine EHRs for the purposes of diabetes research have been comprehensively outlined in a 2017 review [27]. This study benefitted from a large sample size drawn from a UK primary care database of over 15 million individuals, known to be representative of the UK population with respect to age, sex, and ethnicity [17]. This allowed for well-powered comparisons between the three main ethnic groups in the UK. The sample size of this study was significantly larger than those used in other recent UK and European studies; our study included over 160,000 individuals, compared with 2,500 to 24,000 reported elsewhere [23,24,28].

**Table 3. Ethnic differences in therapeutic inertia (failure to intensify treatment within 12 months of HbA1c >7.5%).**

| Treatment stage | Ethic group | N with any HbA1c >7.5% and 12 months' follow-up | Percent experiencing treatment inertia at 12 months, % (n) | Months between first HbA1c >7.5% and intensification/end f-up, mean (SD) | Odds of treatment inertia |
|---|---|---|---|---|---|
| Initiation of noninsulin monotherapy | White | 34,546 | 18.4 (6,354) | 6.9 (13.8) | 1 |
| | South Asian | 1,683 | 16.2 (273) | 6.3 (13) | 0.97 (0.76–1.24), 0.796 |
| | Black | 560 | 17.5 (98) | 6.1 (13.2) | 0.94 (0.69–1.28), 0.683 |
| Intensification to noninsulin combination therapy | White | 54,307 | 67.1 (36,431) | 29.2 (30) | 1 |
| | South Asian | 3,076 | 69.4 (2,135) | 29.8 (28.9) | 1.45 (1.23–1.70), <0.001 |
| | Black | 1,029 | 68.7 (707) | 31 (31.4) | 1.43 (1.09–1.87), 0.010 |
| Intensification to insulin therapy | White | 36,480 | 93 (33,920) | 54.8 (38.7) | 1 |
| | South Asian | 2,061 | 96.2 (1982) | 61.3 (43.1) | 2.68 (1.89–3.80), <0.001 |
| | Black | 702 | 94.4 (663) | 57.8 (42) | 1.82 (1.13–2.92), 0.013 |

Regression model adjusts for age, gender, deprivation, HbA1c value, BMI value, smoking status, macro- and microvascular comorbidities, and depression at start of follow-up, number of consultations, and medications at the start of each follow-up period, calendar year at follow-up start, and clustering by practice. Models for intensification to combination therapy and insulin additionally account for time since diagnosis.

Abbreviations: BMI, body mass index; HbA1c, glycated haemoglobin

Furthermore, the crude time to intensification with insulin reported in our study (3.8 years) matched those of the UK-based cohort studies, which reported median times to insulin initiation between 3.2 and 4.9 years, indicating consistency of data quality and representativeness of the target population [24,28].

Thanks to the quality and outcomes framework, recent improvements in the completeness and quality of both diabetes and ethnicity data have facilitated robust examinations of ethnic differences in the management of type 2 diabetes in primary care settings [19,29]. Diagnoses of T2DM were ascertained using a validated algorithm designed to minimise miscoding and misclassification of diabetes type, reducing the likelihood that individuals with type 1 diabetes or other forms of diabetes were included in our study population [18]. Restriction of the study sample to individuals with at least 12 months of continuous registration prior to their initial diagnosis of T2DM ensured that diagnoses were truly incident and that a sufficient look-back period to capture key baseline covariates was present.

By restricting the analyses to white, South Asian, and black African/Caribbean groups, we were able to make clinically relevant comparisons between well-defined populations with distinct biological, sociocultural, and demographic characteristics, which can be meaningfully characterised as ethnicity [30]. Linkage to area-level deprivation data enabled us to separate the influences of ethnicity and deprivation, which are often conflated when examining health disparities.

General practice characteristics, such as size and participation in local enhanced service schemes, have been found to play a large role in observed variations in the quality of diabetes care [31]. By accounting for the clustering of individuals within general practices, we were able to appropriately account for the influence of practice-level factors on ethnic disparities.

Several limitations may have influenced our findings: Firstly, as EHRs are primarily used for patient care rather than research, data quality and completeness can vary significantly depending on the time period, disease area, and indicator of interest. These issues were

mitigated by using data on a chronic disease condition managed predominantly in primary care settings [32,33] and by adjusting for calendar time in the analysis to account for secular trends in prescribing guidelines and treatment availability.

Secondly, the length of follow-up may have been insufficient to adequately characterise individuals intensifying with insulin therapy. Although 26.5% of the study population intensified from noninsulin monotherapy to noninsulin combination therapy during the follow-up period, only 5.8% of the population underwent all three intensification stages.

Furthermore, our definition of therapeutic inertia may have been too crude when looking at intensification to insulin therapy because individuals may have switched between various noninsulin combinations for some time before up-titrating with insulin therapy. As shown in our descriptive analysis, South Asian and black groups most commonly added noninsulin therapies as their fourth- and fifth-line treatments, whereas white groups most frequently added insulin.

Although the CPRD captures prescriptions made by the general practitioner, it does not hold any information on dispensing data; thus, we cannot know for certain whether prescriptions were filled or taken by the individual. Finally, we were unable to explore ethnic differences in rates of nonattendance to planned appointments, which may be related to ethnic differences in adherence to recommended therapeutic regimes and compliance with diabetes management plans.

## Implications for clinicians and policymakers

The findings of this study highlight clear ethnic disparities in the long-term therapeutic management of type 2 diabetes, which likely contribute, at least in part, to the worse outcomes seen in ethnic minority populations in the UK. Given that baseline HbA1c prior to treatment initiation was higher in the ethnic minority than the white groups, our finding of faster initiation of noninsulin monotherapy in nonwhite groups is commensurate with the more advanced disease at diagnosis in the former groups. It may also reflect a preference for initial pharmacological (as opposed to lifestyle) interventions in ethnic minority groups, possibly indicating greater awareness of the disproportionate risk of major vascular outcomes in South Asian and black populations. Correspondingly, our previous study examining ethnic differences in management of type 2 diabetes around the time of diagnosis found that South Asian and black groups were offered nonpharmacological interventions (structured diabetes management and risk assessments) more quickly than white groups [34].

Reasons for treatment delays can stem from the healthcare system, the practitioner, and the patient. In the UK, these barriers include competing demands on practitioner time, financial constraints of the NHS (particularly in relation to the costs of newer medications), patient adherence, and concerns over side effects [35]. Intensification to insulin is particularly challenging because of the complexity of administration, the level of instruction required, and patient concerns around the use of injectable treatments [36]. Such challenges may be further exacerbated by cultural and language barriers on both the patient [37,38] and provider side [39], potentially explaining excessive delays in treatment intensification among ethnic minority groups. Ethnic differences in health beliefs and attitudes toward medication may also play a role in therapeutic inertia. A recent study from the United States found that statin undertreatment among African American groups was partially explained by lower levels of trust in healthcare practitioners and lower perceived safety of statins in the African American population compared with the white population [40]. Similarly, a UK survey of the Bangladeshi population found that refusal of insulin treatment was associated with fear of premature death, fear of weight gain, loss of independence, and lack of perceived improvements to quality of life [38].

Considering the wider spectrum of cardiometabolic disease, therapeutic inertia remains a substantial problem in the management of blood pressure and lipids. As with diabetes, causes for inertia in other disease areas are multifactorial, stemming from patient and provider, healthcare system, and policy factors. In addition to the important role of patient beliefs and understanding of health conditions and treatment options, other studies have highlighted the importance of education around clinical guidelines for both clinicians and patients, the use of multidisciplinary clinical teams, and the importance of quality improvement initiatives such as pay-for-performance schemes [41,42].

A final consideration is that treatment inertia may be appropriate for certain populations. As treatment decisions are increasingly made jointly between individuals and their care providers, purposeful therapeutic inertia may reflect shared concerns around frailty, treatment burden, and competing health-related priorities [43,44]. Risk of hypoglycaemia increases significantly with age and may be a key consideration in delaying treatment intensification in older age groups [45]. It is likely that some of the delays evident in our study population may be the result of individualised target setting to avoid overtreatment, particularly around the time of insulin intensification. However, even after accounting for burden of comorbidities, age, HbA1c, BMI, and medications, we found compelling evidence that ethnic minority groups nevertheless experience treatment inertia to a far greater degree than the white population.

## Unanswered questions and future research

The first question arising from this work is to determine the relationship between prescribing patterns, long-term glycaemic control, and ethnic differences in micro- and macrovascular outcomes. Secondly, though difficult to quantify in EHRs, determining ethnic differences in adherence to diabetes treatment may shed light on how best to support specific population groups in maintaining good glycaemic control over the longer term. Thirdly, examining ethnic differences in the comparative effectiveness of different treatment regimes with respect to cardiovascular outcomes will be key to developing tailored treatment guidelines for multiethnic populations. Replicating and extending trials such as these to determine whether the risks and benefits of these treatments manifest differently between ethnic groups in real-world settings will form essential next steps toward the personalisation of care in populations with type 2 diabetes.

## Transparency statement

Rohini Mathur is the manuscript's guarantor. She affirms that the manuscript is an honest, accurate, and transparent account of the study being reported; that no important aspects of the study have been omitted; and that any discrepancies from the study as originally planned have been explained.

## Related manuscripts

The authors do not have related or duplicate manuscripts under consideration or accepted for publication elsewhere.

## Supporting information

**S1 RECORD Checklist. RECORD checklist.** RECORD, Reporting of Studies Conducted Using Observational Routinely Collected Data.
(DOCX)

**S1 Fig. Algorithm to assign ethnicity to study participants.**
(TIFF)

**S2 Fig. Directed acyclic graph of the hypothesised relationship between ethnicity, therapeutic inertia, and intermediate confounders.**
(TIFF)

**S3 Fig. Cumulative survival plots for survival analysis by ethnic group.**
(TIFF)

**S4 Fig. Population flowchart for analysis of therapeutic inertia.**
(TIFF)

**S1 Text. ISAC scientific protocol.** ISAC, Independent Scientific Advisory Committee.
(DOCX)

**S1 Table. Code lists for all study variables.**
(DOCX)

**S2 Table. Ethnic breakdown of study population according to the 16 categories of the UK census.**
(DOCX)

**S3 Table. Baseline characteristics for mixed/other and unknown groups compared with white.**
(DOCX)

**S4 Table. Time to diabetes treatment initiation and intensification for white versus other, mixed, and unknown ethnic groups.**
(DOCX)

**S5 Table. Therapeutic inertia at initiation of noninsulin monotherapy using a cutoff of 6.5% for definition of raised HbA1c.** HbA1c, glycated haemoglobin.
(DOCX)

**S6 Table. Ethnic breakdown of individuals included and excluded from analysis of therapeutic inertia.**
(DOCX)

**S7 Table. Full models for time to initiation/intensification and therapeutic inertia.**
(DOCX)

## Author Contributions

**Conceptualization:** Rohini Mathur, Nish Chaturvedi, Liam Smeeth.

**Data curation:** Rohini Mathur.

**Formal analysis:** Rohini Mathur.

**Funding acquisition:** Rohini Mathur.

**Investigation:** Rohini Mathur.

**Methodology:** Rohini Mathur, Ruth E. Farmer, Sophie V. Eastwood, Ian Douglas.

**Project administration:** Rohini Mathur.

**Supervision:** Liam Smeeth.

**Writing – original draft:** Rohini Mathur.

**Writing – review & editing:** Rohini Mathur, Ruth E. Farmer, Sophie V. Eastwood, Nish Chaturvedi, Ian Douglas, Liam Smeeth.

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
