## [Decision Letter · Decision Letter 0]

29 Dec 2019

Dear Dr. Mathur,

Thank you very much for submitting your manuscript "Ethnic differences in initiation and intensification of antidiabetic therapy and therapeutic inertia in adults with type 2 diabetes: A cohort study in the UK Clinical Practice Research Datalink" (PMEDICINE-D-19-04094) for consideration at PLOS Medicine. 

Your paper was discussed among the editorial team and sent to independent reviewers, including a statistical reviewer. The reviews are appended at the bottom of this email and any accompanying reviewer attachments can be seen via the link below:

[LINK]

In light of these reviews, we will not be able to accept the manuscript for publication in the journal in its current form, but we would like to invite you to submit a revised version that fully addresses the reviewers' and editors' comments. You will appreciate that we cannot make a decision about publication until we have seen the revised manuscript and your response, and we expect to seek re-review by one or more of the reviewers. 

We hope to receive your revised manuscript by Jan 17 2020 11:59PM. Please email us (plosmedicine@plos.org) if you have any questions or concerns.

Please let me know if you have any questions. Otherwise, we look forward to receiving your revised manuscript soon. 

Sincerely,

Richard Turner, PhD

rturner@plos.org

Please confirm that author REF was not an employee of Boehringer when the study was being carried out, and that the company had no involvement in planning or carrying out the study. 

Please add the start and end years of the study to your title. 

To the "methods and findings" subsection of your abstract, please add additional demographic information on study participants: i.e., mean age and proportion male. 

Please add a few words to the abstract to summarize the approach to adjustment. 

We ask you to add p values alongside 95% CI where available. 

Please remove elements of repetition from the presentation of your findings, e.g., "... Treatment inertia was higher in south Asian and Black groups at this stage relative to White participants (south Asian OR 1.72 ...; Black OR 1.61 ...)".

Please add a new final sentence to the "methods and findings" subsection of your abstract to summarize the study's main limitations. 

After the abstract, we ask you to add a new and accessible "author summary" section in non-identical prose. You may find it helpful to consult one or two recent PLOS Medicine research papers to get a sense of the preferred style.

Early in the methods section of your main text, please state whether the study had a protocol or prespecified analysis plan (and if so attach the document[s] as supplementary files, referred to in the text). Please highlight analyses that were not prespecified. 

Please move the statement on ethics approval to the methods section of the main text. 

Please avoid claims of "the first" (e.g., in the first paragraph of the discussion section), and where necessary add "to our knowledge" or similar. 

In the discussion section of your main text, please restructure the text to create a discrete paragraph discussing limitations and add "This study had some limitations ..." or similar at the start. 

Throughout your text, please reformat reference call-outs as follows: "... ethnic origin [1-3].".

Please substitute "sex" for "gender" as appropriate, throughout the paper. 

Please check all reference citations to ensure that they match journal style. Several references appear to need additional access information (e.g., reference 33); reference 31 seems to lack a journal name; reference 8 needs reformatting. Please remove all iterations of "[Internet]".

Please revise the attached RECORD checklist so that individual items are referred to by section (e.g., "Methods") and paragraph number rather than by page or line numbers, as the latter generally change in the event of publication. 

Please refer to the checklist in the methods section of your main text. 

Comments from the reviewers:

*** Reviewer #1: 

Thanks for the opportunity to review your manuscript. My role is as a statistical reviewer so my comments are focused on the study design, data, and analysis (and the presentation of these). I appreciate that detailed appendices on the derivation of the ethnicity information and the types of pharmacotherapy (and the other exposure information). This had headed off many of my usual questions I have from studies using GPRD and other similar routinely collected data sources, and will prove to be useful to anyone else attempting to do something similar to your own study. This study uses UK routinely collected data to estimate whether initiation and intensification of pharmacotherapy for T2DM differs according to ethnicity. From my own experience doing similar work I appreciate how complicated an analysis like this is, and I think that you have presented the details of this very clearly. I have some general queries around the study, and some more specific queries and points following that.

General

Was a study protocol or statistical analysis plan developed for this study? It would helpful to see this if it was develop a-priori. 

Will diagnoses and pharmacotherapy provided by specialists (endocrinologists) appear in the dataset, and medicine given during hospital stays?

One of the key parts of the data that this study hinges on is accurate identification of patient's ethnicity, and the algorithm used to do this is clearly presented. This seems reasonable, and rather than give you 'death by sensitivity analyses' asking for the main results to be repeated by the various assumptions that could be made here, is it possible to quantify what the frequency of ethnicity in the study would change if a stricter definition was used where all records had to be consistent? Has there been any methods papers on the UK CPRD looking at the quality of the use of this variable? My own experience with other sources of routinely collected data is that this can wildly vary - although it certainly can be well recorded and reliable.

There is a fair amount of literature looking at T2DM management in South-East Asians in other countries - in this study these were put together into the 'other' category of ethnicity. Was there an insufficient number of self-identifying south-east Asian persons to consider these in this study? Is it possible to have a table that reports on the frequencies of the persons excluded as 'other' according to the 16 category ethnicity variable in the appendix?

Has the use of self-identification to an ethnic group changed over time in the CPRD? Could you clarify how someone would be classified if they had never had valid record of ethnicity in the data?

Specific queries

P6, Paragraph 2. What are the adjustments to the BMI categories?

P6, Paragraph 3. How are participants who move from one form of therapy to a less intense form of therapy, and then to another form of therapy treated? i.e. Monotherapy to none, none to combination therapy? Is the CPRD based on prescription or dispensing records? 

P7. Paragraph 4. Are non-pharmacotherapy periods considered to be a 'drug regime'? 

P7, Paragraph 5. What were the levels in the multilevel Cox model? Is this to account for multiple treatment periods (i.e. none -> mono, mono -> combination) within the same individual? [I've noticed this is clarified below, but it would be helpful to explain the levels here as well] Were proportional hazards between levels of covariates checked, and how was this done? 

P8, Paragraph 3. If a covariate (e.g. microvascular complications) changes during a treatment period, how is this accounted for?

P8, Paragraph 4. Were there any checks of overdispersion with the Poisson regression model?

P11, Paragraph 2/3. I'd also like to see cumulative incidence plots of this if possible (in an appendix is fine), and the median time to intensification (i.e. the time when 50% of those able to receive intensification received it.

*** Reviewer #2: 

The manuscript by Mathur et al deals with a very relevant issue regarding ethnic differences in the implementation of antidiabetic treatment in real-world practice. The authors used the well-known CPRD database to explore the potential ethnic differences in initiation and intensification of hypoglycemic therapy in type 2 diabetes in the UK. The methodology used is sound and the paper well written and easy to read. The findings are relevant for daily clinical practice and for future research in this field. 

This reviewer would like the authors to address the following issues:

- In the Abstract, under Conclusions, the authors include a statement on delays in treatment intensification (second sentence). Actually, delays may be related to many factors and I would recommend rephrasing the sentence or delete it.

- Under Methods, it is not clear whether the information available on drug use is based on prescription by physicians or on pharmacy dispensation. This is relevant as information on the latter is a better proxy to adherence. In addition, this is an issue that deserves a comment in the discussion. 

- Also, under Methods, please explain the method used to handle missing data.

- In Statistical analysis, section on 'Time to treatment initiation and intensification',

- In the section on Results, first paragraph, the sum of percentages provided sum up to 100.1%. Please , check this. Also, please, check the percentages given under 'Patterns of glucose-lowering treatment'.

- Although the duration of diabetes is most probably not different among groups, this information is of interest for the reader. 

- The authors included the number of consultations which is different between groups. However, it is important to know whether the difference may be due to non-attendance to the planned appointments. Do the authors have information on adherence to planned follow-up consultations. This may be also an indicator of adherence to the therapeutical plan; this may clearly influence the outcomes of the study.

- This reviewer understands that the data analyzed are those that correspond to primary care diabetes management. Do the authors have information on access of the subjects to secondary/tertiary care of the different ethnic groups? Were any subjects managed at this level for their diabetes?

- There are several factors known to affect the outcomes measured in this study, both on the patient's and on the professional's side. One of them is on the subject with diabetes, i.e. depressive disorders; do the authors have data on differences among ethnic groups in the prevalence of this co-morbidity? This would be relevant as a confounding variable. 

- Other confounding variables known to affect the outcomes are cultural differences in perception of the disease, perception of the impact on daily life (especially for insulin) and communication barriers between patients and health care professionals (very likely to affect treatment implementation). These variables have been shown to impact on treatment outcome and probably deserve a comment in the discussion.

- In the Discussion, under 'Unanswered questions and future research', I do not see the reason to specifically mention the CAROLINA trial. Please, explain.

*** Reviewer #3: 

Thank you for the opportunity to review this review this article which uses the UK primary care electronic healthcare records to assess the association between Type 2 diabetes glucose lowering treatment intensification and ethnicity. The article shows that despite similar initial treatment intensification there appears to be substantially more inertia in initiating later glucose lowering therapies in those of Black and South Asian ethnic minority background, associated with reduced follow up and HBA1c monitoring.

The article is well written and addresses a question of some importance with (to my non methodologist eyes) robust analysis. However one factor notably missing is an attempt to understand/explain these differences, or relate them to findings in other disease areas. 

The finding of differences later in care is at odds with that of initial treatment and it would be helpful to consider this further. Within the limitations of the dataset is there scope here to start to shed some light on cause of this variation, that might therefore help target action to address these imbalances? For instance how does this variation alter once reduced follow up/monitoring is taken into account - if a large effect this might point to differences in offering or attending following up being key areas to address, rather than patient or physician willingness to accept or prescribe more intense therapy. Is there scope to assess adherence (at least in terms of prescription pick up)? Poor adherence to existing therapy can be a reason for not adding additional therapy. It may also be informative to break down the effect adjusting for confounders in the same way as other authors in related fields have done (for example see figure 3 of Nanna et al JAMA cardiology 2018 PMID: 29898219). 

On brief search it appears to me that there may be substantial related work in other fields relevant to this exact issue including in management of hypertension and treatment of lipids - it is unlikely that an issue with titrating diabetes therapy exists (or is best addressed) in isolation, so it would be helpful to at least put this in context of some of the work in other conditions in the discussion. It would also be helpful comment on possible explanation and related work - for example differences in statin use by ethnicity appear to be heavily influenced by health beliefs (see Nanna et al article above) and a brief search suggests there has been previous qualitative work on this area in type 2 diabetes which may be relevant - for example there appears to be work around ethnicity and health beliefs relating to insulin initiation that is directly relevant to the findings here (e.g. PMID: 27574375 and references within). The substantial increased use of the new more expensive glucose lowering agents in later stages in those of non white ethnicity may be relevant to this. 

Additional points 

1. It is a little confusing exactly what has been adjusted for, and why. Baseline covariates (page 6) suggests factors like smoking and BMI were included as covariates in analysis, but many of these are absent from the related statement of a-priori confounders on page 8. Page 8 suggests to my mind that consultations and number of measurements were also adjusted for, but this does not appear to be the case in table 2. In table 3 all the things on page 6 again crop up. This leaves me rather lost, and also confused why different covariates may be considered (if indeed this is the case) for very closely related analysis of time to intensification of therapy and treatment inertia, which would appear to be aspect of the same thing. Related to this the DAG suggests the authors consider there is a similar relationships for hbA1c and BMI/smoking etc but (if page 8/table 2 is correct) these appear to be treated differently in analysis. One would expect BMI in particularly to affect choice of agent (given clinicians and patients may be more reluctant to use medications associated with weight gain such as insulin/sulfonylureas where there is obesity). 

2. There is to my mind a possibility of residual confounding here that may be exacerbated by the use of adjustment for artificial subgroups (as the covariates description on page 6 suggests) rather than using continuous covariates. Was there a reason to split age into deciles rather than adjust for age as a continuous covariate (in the same way as other groups using this dataset have done?), ditto for BMI which is even more crudely categorised - a BMI of 30 and 60 are likely to have different influences on choice of treatment.

3. We know there is marked regional variation in use of newer agents and potentially rural/urban differences in care. There will also be major differences in ethnicity by rural/urban status and region. I assume this will be accounted for by modelling at the practice level, could the authors reassure me that this is the case?

***

[LINK]

---

## [Decision Letter · Decision Letter 1]

5 Mar 2020

Dear Dr. Mathur,

Thank you very much for re-submitting your manuscript "Ethnic differences in initiation and intensification of antidiabetic therapy and therapeutic inertia in adults with type 2 diabetes: A cohort study in the UK Clinical Practice Research Datalink" (PMEDICINE-D-19-04094R1) for consideration at PLOS Medicine.

I have discussed the paper with editorial colleagues, and it was also seen again by three reviewers. I am pleased to tell you that, provided the remaining editorial and production issues are dealt with, we expect to be able to accept the paper for publication in the journal.

[LINK]

Please let me know if you have any questions. Otherwise, we look forward to receiving the revised manuscript shortly. 

Sincerely,

Richard Turner, PhD

rturner@plos.org

Requests from Editors:

We suggest a more compact title: "Ethnic disparities in initiation and intensification of diabetes treatment in adults with type 2 diabetes in the UK, 2004-2017: a cohort study"

Please add a full point as needed after "hypothesized confounders" in the abstract.

Please adapt "... groups were faster to initiate ..." (abstract) to "... non-insulin monotherapy was initiated earlier in south Asian and Black groups ..." or similar. 

In the "Conclusions" subsection of your abstract, please remove the word "strong", bearing in mind the research design, or substitute "persuasive" or similar. 

At the end of the "Introduction" section of your main text, are (b) and (c) different? Please reword as appropriate.

Please refer to the attached "RECORD" checklist at the appropriate point in your methods section. 

Please remove the short section on "patient and public involvement". 

Under "ethnic differences" (results, third paragraph), please correct "p<0.001".

In the first paragraph of the discussion, please adapt "... experienced significant delays" to "were more likely to experience delays" or similar. 

Please avoid the phrase "white majority". 

Please use the term "diabetes treatment", or similar, rather than "antidiabetic therapy" throughout the ms. 

Please substitute "sex" for "gender" throughout the article. 

Please reposition all reference call-outs as follows: "... adverse diabetes outcomes [12,13]. In the UK ...".

In the reference list, please format author names correctly where needed, e.g., references 32, 28 and 40.

Please add journal names where they are missing, e.g., references 18 and 44. 

In table 1, please remove the colour highlighting of "depression".

Comments from Reviewers:

*** Reviewer #1: 

Thank you for the revision and replies to the comments made on the original submission. Overall I consider my original comments to be resolved with the changes to the manuscript in this version and the inclusion of the protocol in the supplementary materials. The change to use continuous covariates (i.e. BMI) as suggested by reviewer 3 is a good change to the manuscript and the presentations of hazard/odds ratios in Table 2/3 highlights the differences between groups well. I had one query relating to the identification of ethnicity prompted by the additional table, and a comment about the figures in the main part of the paper.

Table S1 is very helpful to understand the selection of ethnic groups as the main exposure in the study. The majority of persons haven an 'unknown' ethnicity classification and are excluded from the analysis, which would presumably mean that many of the persons who are actually 'White' are excluded at this this step. Are the 'White' group very similar to those in the 'Unknown' category? i.e. are the differences seen between ethnicities in treatment intensification driven by selection bias on identification as 'White' in the CPRD? My own experiences with routinely-collected data on self-reported CoB suggest that people and health services that elicit this information from patients are likely to be different to those that do not. A table comparing the baseline characteristics of the excluded 'Other' with 'White would resolve whether a select group more likely to receive treatment intensification was used by only selecting the people with a confirmed 'White' ethnicity code. 

Prior to publication, the Figures 2 and 3 should be revised to a higher standard: 1) there is not main axis title, 2) The numbers in Figure 2 are difficult to read against the striped background, and in the rarer intensification patterns they overlap and can't be read, 3) The table in Figure 3 needs some revision so it's clear there are 5 drug reported across the axis (darker gridlines between drug numbers?).

*** Reviewer #2: 

The authors have properly addressed the issues raised by this reviewer

*** Reviewer #3: 

Thank you for fully addressing my comments. I have only one further comment (for information): while there are clear limitations in assessing adherence in CPRD it is possible to assess whether a patient is requesting sufficient repeat prescriptions to cover the prescribed dose, and this measure has been shown to be associated with glycaemic control in CPRD - see Farmer et al Diabetes Care 2016 PMID: 26681714.

***

[LINK]

---

## [Editor Report · Decision Letter 2]

15 Apr 2020

Dear Dr. Mathur, 

On behalf of my colleagues and the academic editor, Dr. Didac Mauricio, I am delighted to inform you that your manuscript entitled "Ethnic disparities in initiation and intensification of diabetes treatment in adults with type 2 diabetes in the UK, 1990-2017: a cohort study" (PMEDICINE-D-19-04094R2) has been accepted for publication in PLOS Medicine. 

PRODUCTION PROCESS

PRESS

PROFILE INFORMATION

Thank you again for submitting the manuscript to PLOS Medicine. We look forward to publishing it. 

Best wishes, 

Richard Turner, PhD

Senior Editor 

PLOS Medicine

plosmedicine.org